# Assessing the Usability, Feasibility, and Engagement in IM FAB, a Functionality-Focused Micro-Intervention to Reduce Eating Disorder Risk

**DOI:** 10.3390/ijerph22111618

**Published:** 2025-10-23

**Authors:** D. Catherine Walker, Mai P. N. Tran, Lauren E. Leavitt, Dena Contreras

**Affiliations:** Department of Psychology, Union College, Schenectady, NY 12308, USA; maitran20050417@gmail.com (M.P.N.T.); leavittl@union.edu (L.E.L.);

**Keywords:** micro-intervention, body functionality, gratitude, digital mental health intervention, eating disorder, body dissatisfaction

## Abstract

Although our society is becoming increasingly reliant on technology, clinical practice has not yet harnessed digital technology to address the widest audience possible to prevent and treat a range of mental health concerns. The present study aimed to contribute to the literature by exploring the usability, feasibility, and engagement in In the Mirror: Functional Appreciated Bodies (IM FAB), an easily disseminable micro-intervention aimed at reducing body image dissatisfaction and eating disorder risk, piloted in a sample of undergraduate women. We evaluated the usability of the intervention’s procedures and prompts, the feasibility of using the IM FAB program as a smartphone app, and participant engagement to best understand how the participants’ experiences can inform future digital mental health intervention development using the same treatment techniques. Two hundred undergraduate women completed three weeks of mirror exposure sessions and received randomly scheduled text-based journaling prompts in the intervening two weeks. They completed a post-experiment questionnaire, which included the Usability Metric for User Experience (UMUX) scale, app-based feasibility questions, and engagement questions. Usability, feasibility, and engagement scores were high. Participants were generally positive, but with some mixed feedback about transitioning IM FAB to a digital mental health format, noting concerns about accountability and engagement if it was self-directed. Participants generally reported that the text journaling supported what they learned during mirror exposures. These insights can guide the future piloting of the IM FAB program as a mobile app with individualized features.

## 1. Introduction

Digital mental health interventions (DMHIs) are increasingly popular as they may address or mitigate mental health concerns in a low-cost, affordable, and scalable manner. Yet, their success hinges on several fundamental factors: usability, feasibility [1], and user engagement. Psychiatric diagnoses such as anxiety disorders, depression, obsessive–compulsive disorders and eating disorders (EDs) are some of the most chronic and disabling psychiatric conditions [2]. Even though evidence-based treatments are available for these diagnoses, most individuals with these disorders never receive appropriate care because of stigma, expense, or unavailability [3].

### 1.1. Need for Digital Mental Health Interventions (DMHIs)

When college students were offered a universal stepped-care prevention program based on a brief ED risk screening instrument, Lipson and colleagues [4] reported low rates of service utilization (5–18%) for the students with the highest ED risk. Students’ main reasons for not enrolling in their assigned prevention program were: questioning a need for counseling/therapy, preferring to deal with issues on their own, and a lack of time [4]. Furthermore, females were more likely to perceive a need for treatment, be diagnosed, and be treated than men for ED symptoms, as were individuals with higher socioeconomic backgrounds [5]. Additionally, many barriers to treatment-seeking and treatment provision are noted for racially/ethnically minoritized individuals [6] and those of lower socioeconomic status, such as lack of health insurance, transportation difficulties, inability to take time off from shift work positions, etc. [7,8], making it much more difficult for these individuals to seek and receive treatment. Treatment for individuals with ED symptoms is expensive and time-consuming; average costs range from USD 12,146 to USD 20,317 per year for varying treatments for bulimia nervosa [9]. Inpatient treatment for patients with anorexia nervosa ranges from USD 7107–USD 133,609 per year [10]. The time-consuming nature of ED treatment also impacts the family members of patients, with time spent ranging from 72–90.6 h per month [11]. Thus, to overcome these barriers, effective interventions may need to be time-limited, independently accessible (i.e., a self-help modality), and, preferably, would be located where at-risk individuals are likely to search, such as via mobile app-based platforms. Given these costs, universal and targeted prevention efforts are likely to save money and time and improve the quality of life for many individuals and their families.

Data from the Pew Research Center reported that in 2023, 97% of young adults (ages 18–29) owned smartphones, with approximately three-quarters of 18–29-year-olds using their smartphones to search for information about health concerns both in the U.S. [12] and Germany [13]. Furthermore, 20% of young Americans were dependent on smartphones for online access (e.g., did not have broadband access at home). Rates of smartphone dependence are even greater among those from racially/ethnically minoritized groups (21% African Americans and 20% Latinos vs. 12% Whites) and those with lower socioeconomic status (27% of those with incomes <USD 30,000 vs. 6% of those with incomes >USD 75,000) [14]. Thus, digital interventions may be the best platform by which to reach at-risk youths, in particular minoritized and lower socioeconomic groups, who may not have access to ED prevention and treatment elsewhere or who may be overlooked by treatment providers [7,8].

Although several ED DHMI’s have been developed and tested for initial efficacy, usability, feasibility, and engagement [15,16,17,18,19], these interventions are largely targeted towards those with active EDs rather than tailored towards reducing risk factors and preventing the development of EDs, save for two [20,21]. A recent review of all ED-related smartphone apps conducted in June 2021 noted that 65 apps supporting ED treatment were on Google Play or the App Store for iOS, with only 13 research articles describing and supporting 4 ED apps [22]. Additionally, the apps that have been tested are not always made publicly available soon after initial support for their efficacy, given the different timelines of academic vs. consumer-facing products. Thus, there is both a need for innovation in this space and empirically-supported apps to be made publicly available, even at the same time as additional research is conducted and the product is refined. In this way, products with empirical support created by experts may have greater prevalence in publicly available online marketplaces compared to available eating disorder and body image intervention DMHIs.

### 1.2. The Need for Usability, Feasibility, and Engagement Research

Digital mental health interventions provide a novel approach to closing this gap between those who would benefit from prevention and treatment and those who actually engage with prevention or treatment, in any format [1,23,24]. Provided through smartphones, websites, wearable sensors, or some combination of these, DMHIs hold out the hope of acceptability, flexibility, and anonymity. Yet, prior to such tools being able to flourish, they need to pass basic tests of usability, feasibility, and engagement—terms that encompass not just navigation and user interface but how easily the content of the intervention can convey the intended messages [1]. Park et al. [1] emphasize that two pillars should come before any efficacy trials of DMHIs: feasibility and acceptability (which includes usability and engagement). They suggest a model in which usability (ease of interaction and navigation by users, whether the technology is appropriate to the user’s life context) and readability (clearness, comprehensibility, tone of written or visual information) are necessary precursors to large-scale and ethical DMHI trials and, ultimately, adoption. An additional pillar, engagement, refers to whether the user can instantly and emotionally comprehend and relate to the content [1].

Langdon et al. [25] assessed a computer-based motivational intervention with patients with substance abuse disorders receiving buprenorphine treatment. They established that the participants liked the program’s simplicity, minimal cognitive load, and tailoring. The concise, stigma-reduced content enabled more emotional engagement, particularly when presented via mobile text messages. Interestingly, participants also cited message tone, frequency, and timing as key usability factors, implying that readability is not just a matter of linguistic complexity but also tone, empathy, and rhythm. Kanuri et al. [26] reported on the feasibility and cultural adaptation of an anxiety-reduction CBT course for Indian college students. Presented in short modules with culturally translated metaphors and Hindi narration, the intervention ranked in the top 10% on usability tests. Qualitative feedback indicated that students appreciated content clarity and navigability but recommended changes in dropdown menu organization and font readability. Kanuri et al. [26] and Langdon et al. [25] provide examples of how DMHIs may be modified and piloted in particular clinical or cultural settings, allowing users to provide feedback on what works and what does not prior to refining and further testing the DMHI. These pilot studies underscore the fact that good intentions and theoretically sound content are not enough if the delivery mechanism is opaque, confusing, or incompatible with the user’s daily life. For example, if a digital intervention faithfully adapts an evidence-based intervention to a digital format, there may be reasons why the evidence-based intervention is less effective in the digital format. User-recommended changes may improve its usability and acceptability, which would, in turn, be expected to improve the efficacy of the intervention. Park et al. [1] contend that prior to developing and broadly testing DHMIs, assessing acceptability and usability are not technicalities but rather ethical imperatives, with skipping usability testing potentially leading to negative outcomes, from user disengagement to the inability to scale.

### 1.3. Current Study

The current study contributes to the growing implementation science literature by exploring the usability, feasibility, and participant engagement of a digital intervention in which individuals appreciate the functions of their bodies in the mirror, called In the Mirror: Functional Appreciated Bodies (IM FAB). IM FAB is an easily disseminable micro-intervention aimed at reducing college-age women’s body image dissatisfaction, with the ultimate goal of reducing ED risk. IM FAB was created to address the lack of digital interventions for body image, especially among at-risk youths who do not have access to traditional interventions. We aimed to evaluate the usability of the intervention’s procedures and prompts, the feasibility of use if it were translated into a fully digital intervention, as well as participant engagement, to best understand how the participants’ experiences can inform future DMHI development using the same treatment techniques. We assessed whether participants thought IM FAB would be of interest to them if it were provided as a smartphone application (feasibility) in the future. Designed to reduce body dissatisfaction and lower ED risk in college women, IM FAB is a novel, brief digital intervention that pairs functionality-focused mirror exposure exercises with text message journaling prompts on gratitude themes. Participants were randomly assigned to a Functionality condition, Active Comparator condition, or Assessment-Only control condition. There were four assessment timepoints: baseline (T1), post-intervention in week 3 (T2), and at both one-month (T3) and four-month (T4) follow-ups after T2. Women in the Functionality condition received audio-guided mirror exposure sessions on how to think about their body, focusing on their body’s physical, sensory, and creative capabilities (see Appendix A for audio-guided mirror exposure scripts). They were asked to perform this in a room in a lab without a researcher present in the room with them. Additionally, Functionality participants received text message prompts that were scheduled in advance to be sent at three random times between 9 a.m. and 9 p.m. every other day in the two weeks between their in-person mirror exposure sessions. These texts were forwarded to the project director (DCW) for monitoring and were accessible via a password-protected website accessible to the research team. Texts asked participants to write about gratitude for different facets of their body’s functionality (e.g., mobility, strength, ability to connect with others) for Functionality participants (see Appendix A for text prompts). In contrast, the Active Comparator condition received appearance-neutral mirror exposure, in which participants were only asked to examine body parts but not told how to think about those body parts (see Appendix A). Like the Functionality condition, Active Comparator participants completed mirror exposure sessions whilst listening to an audio recording in a private room in a laboratory. Active Comparator participants received text prompts unrelated to body image intended to prompt gratitude for other parts of their lives (e.g., relationships, personal growth; see Appendix A). We assessed participants’ functionality appreciation, body appreciation, eating disorder symptoms, body checking, body image avoidance, and multidimensional measures of body image. The randomized controlled trial demonstrated statistically significant improvements in body functionality appreciation and body image satisfaction for the Functionality condition compared to either the Active Comparator or the Assessment-Only control condition. For additional details about the IM FAB randomized controlled trial procedure and results, please see Costello et al. [27] and the Appendix A. The purpose of the current study was to describe the usability, feasibility, and engagement findings for the IM FAB micro-intervention, with particular attention to whether participants would want to complete IM FAB via a mobile phone application. We hypothesized that, similar to other DMHIs, participants would report generally high usability, feasibility, and engagement ratings. There were no specific a priori hypotheses about group differences in usability, feasibility, or engagement due to minimal prior research indicating group differences between these two similar procedures on the constructs of interest; however, we conducted between-groups tests as an exploratory aim. We assessed both active intervention conditions (Functionality and Active Comparator) on the three variables of interest—usability, feasibility, and engagement at T2.

## 2. Materials and Methods

### 2.1. Participants

Participants from the full IM FAB study were 287 female-identifying undergraduate students at a small liberal arts college and a large state university in the northeastern U.S. Participants were not screened for ED or body image dissatisfaction. Participants were selected based on three inclusion criteria: (1) ≥18 years old; (2) self-identified as female; and (3) had never participated in a cognitive-dissonance-based body image group run on the same campus as one of the data collection sites [28]. For the usability, feasibility, and engagement analyses, we only examined those who were randomized to the Functionality (*n* = 110) and Active Comparator conditions (*n* = 90) and completed the post-intervention (T2) assessment, with a *M*(*SD*)_AGE_ = 19.63 (1.30) years and *M*(*SD*)_BMI_ = 24.10 (5.22). Participants were predominantly White (*n* = 122, 61%) and not Latina (*n =* 171, 86.8%). For a full description of recruitment and participant flow, see Costello et al. [19]. See Table 1 for demographic information for participants in the current study.

### 2.2. Measures

#### 2.2.1. Usability

A modified version of the four-item Usability Metric for User Experience (UMUX) [21] was used in the current study to assess usability. The UMUX is rated on a Likert scale from 1 (*Strongly disagree*) to 7 (*Strongly agree*). Participants were given the following information prior to responding to UMUX items: “We are interested in understanding how ‘usable’ the study was for you. In other words, how easy or hard was it to do the things that were asked practically (rather than emotionally)? Were there any frustrating aspects of the study set-up that you would change or fix were we to run a similar study in the future?” Sample items from the modified UMUX include “The study’s procedures and capabilities (mirror sessions & text prompts) meet my requirements” and “I have to spend too much time correcting things with this study’s procedures (mirror sessions & text prompts).” After reverse-scoring two negatively-valenced items, items were averaged, with higher scores reflecting greater usability. We also asked participants an open-ended usability question: “Please provide us with any additional feedback on the ease of use of the study’s procedures.” The UMUX has previously demonstrated reliability and validity [29,30], with a Cronbach’s alpha of 0.69 and McDonald’s omega of 0.66 in the current study.

#### 2.2.2. Feasibility

The research team developed the 5-item feasibility questionnaire using the UMUX as a basis. We used the same 7-point Likert scale from 1 (*Strongly disagree*) to 7 (*Strongly agree*). Participants first read the following instructions: “Feasibility of app-based delivery: We are interested in understanding your opinions on doing a similar program at home using a Smartphone App that prompted you to do the weekly mirror exposures and regularly texted you to prompt you to practice gratitude, with a few short questionnaires being asked directly on the phone app before and after the mirror exposure sessions.” Example items included “I would have preferred doing this program if it were fully app-delivered;” “I would use an app like this in my own life;” and “I would recommend a program like this on an app to a friend or relative.” These questions were followed by an open-ended question: “Please provide us with any additional feedback on how you would feel about doing a similar program on your own via a Smartphone App.” After reverse-scoring one item, items were averaged, with higher scores reflecting greater feasibility for an app-based delivery format. Cronbach’s alpha in the current study was 0.70, and McDonald’s omega was 0.63.

#### 2.2.3. Engagement

The research team developed the 9-item engagement questionnaire using UMUX as a basis. We used the same 7-point Likert scale from 1 (*Strongly disagree*) to 7 (*Strongly agree*). Participants first read the following instructions: “We are interested in understanding how you felt about the study procedures (emotionally, rather than procedurally or practical issues).” Example items included: “I found the mirror exposure engaging or interesting;” “I felt that the mirror exposure was helpful;” “I was bored when responding to the gratitude texts.” After reverse-scoring four items, items were averaged, with higher scores reflecting greater engagement. Cronbach’s alpha in the current study was 0.84, and McDonald’s omega was 0.65.

### 2.3. Procedure and Manipulation Check

Participants were randomized via a random number generator to their condition (Functionality, Active Comparator, or Assessment Only control). After three consecutive weeks of mirror exposure and two weeks completing randomly scheduled text micro-journaling responses, participants completed a post-experiment questionnaire, which included the UMUX scale, app-based delivery feasibility questions, and the engagement questionnaire described above. See Costello et al. [27] for a detailed description of the IM FAB procedure. Participants completed eight true/false manipulation check questions after completing their mirror exposure each time, in addition to five visual analogue scales asking about their experiences during the mirror exposure. These manipulation check items are included in the Appendix A.

### 2.4. Data Analysis

Responses were examined using descriptive statistics and synthesis of open-ended comments. Independent samples *t*-tests were used to examine whether the Functionality and Active Comparator conditions differed across any of the outcome variables.

## 3. Results

### 3.1. Quantitative Responses

Means and standard deviations for each item are presented in Table 2 collapsed across both conditions, given the lack of a priori hypotheses regarding between-group differences. All participants completed all true/false manipulation check questions correctly. However, participants assigned to the Active Comparator condition engaged in behaviors more aligned with Functionality prompts during 36 of 531 total mirror exposure sessions (6.8%). We conducted independent sample *t*-test analyses to see if participants across the Functionality and Active Comparator conditions differed in their perception of the intervention’s usability, feasibility, and engagement.

#### 3.1.1. Usability

UMUX scores were high, *M*(*SD*)_UMUX_ = 6.20 (0.86) overall, with no significant differences, *t*(155) = 0.41, *p* = 0.61, between the Functionality *M*(*SD*)_UMUX_FUNC_ = 6.23 (0.87) and Active Comparator *M*(*SD*)_UMUX_AC_ = 6.17 (0.85) conditions.

#### 3.1.2. Feasibility

Feasibility scores were moderate, *M*(*SD*)_UMUX_ = 4.14 (1.30), with no significant differences, *t*(156) = 0.47, *p* = 0.46, between the Functionality *M*(*SD*)_UMUX_FUNC_ = 4.19 (1.26) and Active Comparator *M*(*SD*)_UMUX_AC_ = 4.09 (1.34) conditions.

#### 3.1.3. Engagement

User engagement scores were high, *M*(*SD*)_UMUX_ = 5.18 (0.98), with no significant differences, *t*(156) = 2.93, *p* = 0.11, between the Functionality *M*(*SD*)_UMUX_FUNC_ = 5.41 (0.85) and Active Comparator *M*(*SD*)_UMUX_AC_ = 4.96 (1.04) conditions. As an additional quantitative marker of engagement, at a reviewer’s suggestion following preregistration, we examined completion rates. Participants were asked to respond to six text prompts. Each text above a pre-set character limit was sent as additional texts, such that six or more responses means participants completed all text prompts: 61.24% of participants completed all six responses. See Figure 1 for participants’ frequency of responding.

### 3.2. Qualitative Responses

Data were inferred from participants’ responses to open-ended questions. Ease of use was reported by participants, and most liked the repetitive format. For feasibility, most recommended app-based delivery due to convenience, discretion, and notification capability. Readability was discussed in 64% of qualitative responses. Positive tone and readability in the style of a text message were praised by users (*n* = 9). Shared among many were the following themes: “positive and simple wording,” “not overwhelming,” and “easy to understand, even when emotional.” Some users commented that longer or more abstract prompts (e.g., more philosophical gratitude texts) were less readable. Engagement ratings indicated moderate-to-high emotional engagement. It was reported that mirror exposure was uncomfortable but effective. People in the Functionality condition interacted with mirror exposure as more meaningful than those in the Active Comparator condition. Others reported more body checking, in line with the requirement for content warnings for self-paced release. Open-ended feedback from participants offered further insight into their emotional process and personal reflection. Many characterized the experience as “eye-opening” or “surprisingly emotional,” and some explained how the daily prompts facilitated more deliberate reflection. For example, one participant wrote, “the texts made me reflect on my legs not for their appearance but for how they move me through my day.” Another wrote, “the mirror exercises were difficult at first, but they began to make me less ashamed.” However, participants also noted they would have felt hesitant were others in the room (others were not in the room for the in-person mirror exposure but may have been if participants completed the assignment in their dorm rooms, for example) and not being intrinsically motivated to engage in the IM FAB program if it were solely app-delivered and not part of an incentivized randomized controlled trial. It is noteworthy that some participants reported that they did not have body image concerns, but that it may be more effective or motivating for those who do. As such, it is likely that some contextual factors exist (e.g., classes, roommates, time) that affected the perceived usability and feasibility of the intervention, even for those who were more intrinsically motivated.

Separating responses regarding their engagement if the program were delivered via smartphone app, there were 21 positive responses, 10 negative responses, and 15 neutral or ambivalent responses that contained both positively- and negatively-valenced content. All qualitative responses are presented in a table in the Appendix A, categorized as positive, neutral, and negative. These qualitative findings demonstrate the profound, embodied impact of even short-term interventions if crafted thoughtfully to emotional and textual readability, with likely greater impact for those higher in pre-existing body dissatisfaction, as has been previously reported [31].

### 3.3. Main Outcomes of the Randomized Controlled Trial

We also measured body appreciation, functionality appreciation, appearance evaluation, physical functionality orientation, body checking, body image avoidance, appearance orientation, and eating disorder symptoms in a separate paper presenting the main outcomes of the randomized controlled IM FAB trial [27]. Using multilevel modeling, we found that the Functionality condition demonstrated significant improvements relative to the Active Comparator condition in functionality appreciation at T2 and T4, body appreciation at T2, and eating disorder symptoms at T2. Further, the Functionality condition exhibited significant improvements compared to the Assessment Only condition in functional appreciation at T2, appearance evaluation at T3 and T4, and body checking at T3. No other comparisons were consistently significantly different. Thus, functionality-focused ME in this study may be a useful micro-intervention to improve positive body image. However, an in-depth discussion of the intervention and of the main outcome findings are beyond the scope of this paper. The current paper sought to address the usability, feasibility, and engagement of the IM FAB procedure, which was delivered in person/on Zoom for mirror exposure and via text for the intervening weeks, as it was, and if it were to be delivered as a solely app-based intervention.

## 4. Conclusions

The current study aimed to determine the usability, feasibility, and user engagement of the IM FAB procedure, an intervention that is not yet a smartphone application, for potential development as a smartphone application. Similar to prior research, both the IM FAB Functionality condition and the Active Comparator condition demonstrated that usability (study’s procedures: mirror exposure and text prompts), feasibility for app-based delivery (degree to which they would have liked participating in IM FAB on a smartphone app), and engagement (degree to which the mirror exposure and texts were interesting or engaging) were positively rated, supporting the importance of this initial step prior to intervention development in a digital format and widescale delivery. Open-ended feedback allowed us to qualify interpretation and suggested that while the mirror exposures were challenging, they generally were helpful in reframing participants’ experiences of their body over time. While some participants thought an app would be particularly useful for a behavior and experience that is generally private (examining and thinking about their body), some noted they may be less likely to use the app if it was not part of a research study with higher accountability. Thus, any app developed would need to build in either gamification features or accountability features to help keep users engaged. We recommend including features such as gamification, where participants can earn stars, connect with friends via their contacts, share stars with friends, create streaks for using the app each day, congratulate friends for completing streaks, allow them to refill a missed day by completing an additional activity, show participants the testimonies from prior participants of the benefits of continuing the activities on the app every so often, and provide notifications to remind users to sign back in every so often, as examples. We used a unicorn as a “mascot” for the trial, so we envision that users could have a unicorn “mascot” that they start out with who is in grey scale, and they could see their unicorn become more vibrant and colorful and develop new features (e.g., wings, glitter, etc.) the more that users engage with the app.

Additionally, the usability, feasibility, and engagement testing enabled us to see that participants generally reported that the text journaling supported what they were learning in the mirror exposures, facilitating more deliberate reflection of those exposure activities. Most intervention programs do not test each small portion of the intervention separately due to the logistical challenges of attempting this. However, the qualitative feedback suggests that pairing these mirror exposures with text journaling may help users reinforce and support the effects of the mirror exposure, as expected based on cognitive behavioral theory [32].

The study should be considered alongside its limitations. The study sample was limited in terms of demographic diversity, as participants were generally from the northeastern U.S., were obtaining a four-year college degree, and were of a similar age. The study also only included female participants. The results likely cannot be generalized to demographics aside from undergraduate women in the U.S. We could not evaluate how demographic factors affect the effectiveness of the IM FAB program due to a lack of statistical power to examine differences between multiple groups with low sample sizes. Thus, similar research is needed in more geographically, racially/ethnically, and socioeconomically diverse individuals across genders. The study also did not pre-screen participants with EDs, meaning that the results are only suggested to be a potential prevention tool for those at risk of developing EDs. Additionally, the researchers developed their own feasibility and engagement questionnaires to fit with the aims of the current study. Although the internal consistency was supported in the current sample, it is optimal to use validated measures when they exist. Some of the items, in particular, *feasibility* item 3 (“I would probably not have actually done the mirror exposures at home if the study was delivered via an app.”) had a high standard deviation (2.13 on a 1–7 scale), indicating substantial variability within our sample and high skewness and kurtosis in UMUX item 4. Nonetheless, variability, skewness, and kurtosis were all acceptable for the total scales, suggesting appropriateness for analyses as a global scale [33]. Given the low number of items, especially for the UMUX, it is likely that coefficient alpha represents a more accurate estimate of the reliability of the scale than omega [34], and some researchers have noted concerns with the often-used rule of thumb of a 0.70 cut-off [35] and that alpha is often an underestimate of reliability. Nonetheless, higher alpha and omega coefficients generally suggest higher internal consistency, and reliability is a necessary precondition for validity. We have yet to confirm the scales’ test–retest reliability, content validity, and construct validity, all of which are crucial next steps, as is confirming its internal consistency. Furthermore, although we assessed socioeconomic status via parental education, we did not have sufficient power to determine whether there were group differences across socioeconomic classes in the current study. Lastly, the current study was conducted in person, with participants completing mirror exposure in a private room, following guided audio instructions, and receiving text prompts during the week between mirror exposure sessions prior to COVID, and mirror exposure sessions were conducted on Zoom during COVID. Thus, participants were receiving components of a micro-intervention that could be easily converted to a smartphone app, but they were not actually participating in IM FAB on a smartphone app. Thus, participants’ actual experiences using an app may have differed from their expected experiences. One important way in which this is likely the case is user engagement. Participants in our highly controlled randomized controlled trial were paid for completion of each part of the study. Thus, user engagement in real-world conditions would likely differ significantly. If IM FAB is developed into a mobile smartphone application, collecting engagement data without financially incentivizing participants to use the app is an important future direction to better understand and improve its user engagement over time.

In addition to addressing the limitations of the current study, future directions include piloting the IM FAB program as a mobile application with individualized modules, increasingly intuitive pacing tools for mirror exposure, and interactive elements to facilitate emotional processing and engagement. Additional studies assessing IM FAB should take into account other readability indicators, such as health literacy measures and recall of understanding, to better place the self-report data into context. Future research should also explore motivational factors that would encourage more usage of IM FAB, such as those noted previously (e.g., unicorn mascot who develops over time, connection with friends, keeping a streak by engaging, notification features, etc.) and with the ability to carry out exposure on the photo feature of the app or to carry out an at-home mirror exposure. Finally, studies can evaluate the extent of the benefits that the inclusion of gratitude journaling will bring to participants with EDs and body image dissatisfaction, regardless of the forms of interventions.

Additionally, longitudinal follow-up on user engagement, maintenance of behavior change, and course of symptoms can more effectively place the value of micro-interventions in a temporal context. Use of user-centered design throughout the development process and pilot testing with diverse cultural, linguistic, and socioeconomic populations will be critical in ascertaining the equitable benefit and broad uptake of digital technologies such as IM FAB.

## Figures and Tables

**Figure 1 ijerph-22-01618-f001:**
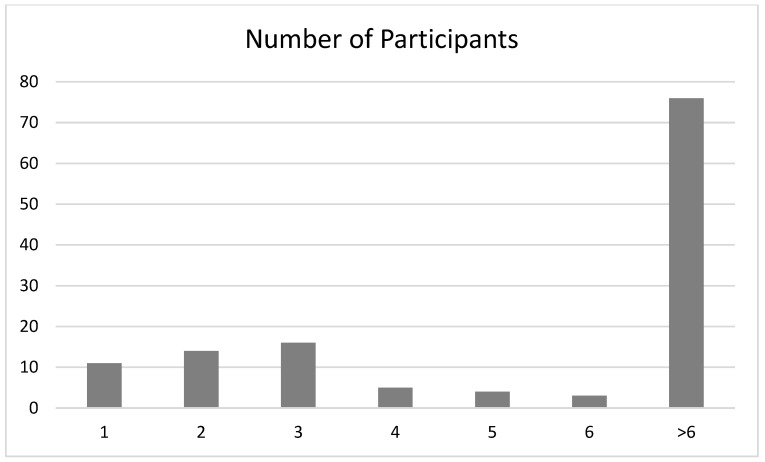
Proportion of participants who complete all six text prompts.

**Table 1 ijerph-22-01618-t001:** Demographics information for the sample (*n* = 200).

Demographics Characteristic	Functionality Condition (*n* = 110)	Active Comparator (*n* = 90)
	*M* (*SD*)	*M* (*SD*)
BMI	24.2 (5.20)	23.96 (5.26)
Age	19.55 (1.28)	19.73 (1.32)
Race	*n* (%)	*n* (%)
Native American	0 (0.00)	0 (0.00)
Asian	20 (18.18)	11 (12.22)
Black	14 (12.72)	16 (17.78)
Hawai’ian/Pacific Islander	1 (0.91)	0 (0.00)
White	67 (60.91)	55 (61.11)
Other	3 (2.73)	4 (4.44)
Multiracial	5 (4.55)	4 (4.44)
Ethnicity		
Latina	16 (14.55)	10 (11.11)
Non-Latina	93 (84.55)	78 (86.67)
Prefer not to answer	1 (0.91)	2 (2.22)
Mother’s Highest Education		
Some High School	5 (4.55)	5 (5.56)
High School Degree	11 (10.00)	9 (10.00)
Some College	11 (10.00)	9 (10.00)
Associate’s Degree	12 (10.91)	6 (6.67)
Bachelor’s Degree	42 (38.18)	22 (24.44)
Some Graduate Coursework	2 (1.82)	0 (0.00)
Graduate Degree	26 (23.64)	38 (42.22)
Did Not Respond	1 (0.91)	1 (1.11)
Father’s Highest Education		
Some High School	9 (8.18)	8 (8.89)
High School Degree	16 (14.55)	9 (10.00)
Some College	6 (5.55)	6 (6.67)
Associate’s Degree	14 (12.72)	9 (10.00)
Bachelor’s Degree	30 (27.27)	28 (31.11)
Some Graduate Coursework	2 (1.82)	1 (1.11)
Graduate Degree	30 (27.27)	26 (28.89)
Did Not Respond	3 (2.73)	3 (3.33)

All *p*’s > 0.05 (independent samples *t*-tests for continuous variables and chi-square analyses for categorical variables).

**Table 2 ijerph-22-01618-t002:** Response information to measures.

Items	Mean (SD)	Skew	Kurtosis
Usability			
1. The study’s procedures and capabilities (mirror sessions & text prompts) meet my requirements.	6.17 (1.08)	−1.87 (0.19)	0.39 (0.39)
2. The study’s procedures (mirror sessions & text prompts) are a frustrating experience. (R)	5.83 (1.48)	−1.34 (0.19)	1.00 (0.39)
3. The study’s procedures (mirror sessions & text prompts) are easy to use.	6.56 (0.78)	−1.87 (0.19)	0.39 (0.39)
4. I have to spend too much time correcting things with this study’s procedures (mirror sessions & text prompts). (R)	6.25 (1.29)	−2.02 (0.19)	3.62 (0.39)
Mean UMUX	6.19 (0.89)	−0.98 (0.19)	−0.11 (0.37)
Feasibility			
1. I would have preferred doing this program if it were fully app-delivered.	3.69 (1.8)	0.08 (0.19)	−0.84 (0.38)
2. I would have preferred doing the mirror exposure at home if it was on an app.	3.74 (1.92)	0.09 (0.19)	−1.09 (0.39)
3. I would probably not have actually done the mirror exposures at home if the study was delivered via an app. (R)	4.59 (2.13)	−0.47 (0.19)	−1.20 (0.38)
4. I would use an app like this in my own life.	4.04 (1.94)	−0.14 (0.19)	−1.23 (0.38)
5. I would recommend a program like this on an app to a friend or relative.	4.62 (1.82)	−0.42 (0.19)	−0.87 (0.38)
Mean Feasibility	4.14 (1.30)	−0.06 (0.18)	−0.11 (0.37)
Engagement			
1. I found the mirror exposure engaging or interesting.	5.16 (1.46)	−0.51 (0.19)	−0.66 (0.38)
2. I found the texting assignments engaging or interesting.	5.18 (1.47)	−0.84 (0.19)	−0.31 (0.39)
3. I thought that the mirror exposure was uncomfortable and distressing. (R)	5.06 (1.60)	−0.48 (0.19)	−0.76 (0.38)
4. I felt that the mirror exposure was helpful.	5.11 (1.35)	−0.51 (0.19)	−0.31 (0.39)
5. I thought that the gratitude texts were uncomfortable and distressing. (R)	5.88 (1.36)	−1.40 (0.19)	1.61 (0.39)
6. I felt that the gratitude texts were helpful.	5.29 (1.46)	−0.84 (0.19)	−0.39 (0.39)
7. I would recommend this study (or doing the same mirror exposure and gratitude texting) to a friend or relative.	5.41 (1.41)	0.01 (0.19)	−1.01 (0.38)
8. I was bored during the mirror exposure sessions. (R)	4.13 (1.79)	−0.73 (0.19)	−0.07 (0.39)
9. I was bored when responding to the gratitude texts. (R)	5.41 (1.41)	0.73 (0.19)	−0.07 (0.39)
Mean Engagement	5.14 (0.97)	−0.026 (0.18)	0.07 (0.37)

## Data Availability

The data that support the findings of this study will be posted on osf.io upon publication of the manuscript and were pre-registered at https://osf.io/4jfp2/?view_only=5d8aeb391c9a4a0495465ab3bd6c5b4d (accessed on 20 August 2025).

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
