# Peer review of "Assessing the Usability, Feasibility, and Engagement in IM FAB, a Functionality-Focused Micro-Intervention to Reduce Eating Disorder Risk"

_ijerph, 2025, doi:10.3390/ijerph22111618_

Round 1

Reviewer 1 Report

Comments and Suggestions for Authors

Thank you for giving me the opportunity for reviewing this paper. The work tackles a significant and current problem at the nexus of body image therapies and digital mental health. The emphasis on a micro-intervention's acceptability and usability (IM FAB) offers helpful information on whether it is feasible to convert well-established therapeutic methods into a scalable, technology-based format. The manuscript might use more tightening, elaboration, and explanation in a few places, but overall it is clear and well-motivated.

Major Comments

Controls and study design

The evaluation of the intervention's distinct contribution is limited by the lack of a control or comparison condition. Is it possible to interpret usability/engagement data in relation to other digital health interventions?

Clarifying whether or whether outcomes other than usability (such as eating disorder risk or body image dissatisfaction) were measured would be beneficial. The paper would benefit from even basic descriptive data.

Limitations of the sample

Only female undergraduates are included in the sample, making it a limited demographic. In the issue of generalizability, this constraint ought to be underlined further.

Were there differences in age, race, or body image issues? Demographic information would provide important background.

Level of feedback

It seems like the discussion of conflicting participant comments is not well-developed. Could the writers go into more detail regarding certain issues with involvement and accountability (such as a lack of reminders, a loss of social support, or a decline in motivation)?

Were there any variations in the three weeks' engagement patterns (e.g., peak engagement, drop-off rates) that could help guide the design of the app?

Delivery and fidelity of interventions

The organization and oversight of the mirror exposure sessions require further explanation. Did participants get any kind of guidance, or was adherence solely self-reported?

In a similar vein, how were journaling prompts sent and monitored? For app translation, it is essential to comprehend fidelity.

Directions for the future

Individualized features for upcoming app versions are mentioned in the end. It would be helpful to indicate which qualities the authors suggest or which participants asked for (e.g.,Reminders, gamification, progress monitoring, and personalization)

Minor Remarks

Making a stronger connection between the possibility for IM FAB and digital technological deficiencies could tighten the introduction.

To make it easier for readers to follow findings across domains, think about rearranging the data to more clearly distinguish between usability, feasibility, and engagement.

The abbreviation "IM FAB" is introduced suddenly. The name's brief justification (such as "Function Appreciated Bodies") ought to come first.

There is repetition in some of the wording (for example, "usability, feasibility, and engagement scores were excellent" might be more nuanced).

This study's standing in the literature would be improved by citations to relevant digital mental health interventions.

Author Response

Reviewer 1

Thank you for giving me the opportunity for reviewing this paper. The work tackles a significant and current problem at the nexus of body image therapies and digital mental health. The emphasis on a micro-intervention's acceptability and usability (IM FAB) offers helpful information on whether it is feasible to convert well-established therapeutic methods into a scalable, technology-based format. The manuscript might use more tightening, elaboration, and explanation in a few places, but overall it is clear and well-motivated.

We thank Reviewer 1 for their kind words and their time reviewing the manuscript! We have addressed each of Reviewer 1’s comments below.

Major Comments

Controls and study design

The evaluation of the intervention's distinct contribution is limited by the lack of a control or comparison condition. Is it possible to interpret usability/engagement data in relation to other digital health interventions?

We thank the Reviewer for the suggestion. Our study had a control condition, which was the Assessment Only group. In this control group, participants were not given any treatment, and were assessed at T1, T2, T3, and T4. However, because they received no intervention, they did not receive questions pertaining to the usability, feasibility of app-based delivery, and engagement in the program that we gave at T2. We did not mention the control group in this paper because it was not relevant to what we were examining in this paper. We have added this information in the procedures section, as follows.

On lines 128-130 in the original manuscript, we noted the presence of a control condition here: “Participants were randomly assigned to a Functionality condition, Active Comparator, or Assessment Only control condition.”

We added: “We assessed both active intervention groups (Functionality and Active Comparator) on the three variables of interest — usability, feasibility, and engagement.”

Clarifying whether or whether outcomes other than usability (such as eating disorder risk or body image dissatisfaction) were measured would be beneficial. The paper would benefit from even basic descriptive data.

We thank the Reviewer for the feedback. We have also measured the mentioned variables, but the scope of this paper resides mainly in addressing the extent to which IM FAB is usable, feasible, and engaging, so we had originally omitted it from this paper and referred readers to the paper that presented primary outcome results. Nonetheless, we believe the Reviewer brought up a question that most readers will have, so we have added more information on the primary outcomes in the introduction and results sections, and then referred readers to the primary outcome paper, as follows:

Introduction:

“Women in the Functionality condition received audio guided mirror exposure sessions on how to think about their body, focusing on their body’s physical, sensory, and creative capabilities. They were asked to do this in a room in a lab without a researcher present in the room with them. Additionally, Functionality participants received text message prompts that were scheduled in advance to be sent at three random times between 9 AM and 9 PM every other day in the two weeks between participants’ in-person mirror exposure sessions. These texts forwarded to the project director (DCW) for monitoring and were accessible via a password-protected website accessible to the research team. Texts asked participants to write about gratitude for different facets of their body’s functionality (e.g., mobility, strength, ability to connect with others) for Functionality participants. In contrast, the Active Comparator condition received appearance-neutral mirror exposure, in which participants were only asked to examine body parts but not told how to think about those body parts. Like the Functionality condition, Active Comparator participants completed mirror exposure sessions listening to an audio recording in a private room in a laboratory. Active Comparator participants received text prompts unrelated to body image intended to prompt gratitude for other parts of their lives (e.g., relationships, personal growth). We assessed participants’ functionality appreciation, body appreciation, eating disorder symptoms, body checking, body image avoidance, and multidimensional measures of body image. The randomized controlled trial demonstrated statistically significant improvements in body functionality appreciation and body image satisfaction for the Functionality condition compared to either the Active Comparator or an assessment-only control condition. For additional details about the IM FAB randomized controlled trial procedure and results, please see Costello et al., [27].”

Results:

“3.3. Main Outcomes of the Randomized Controlled Trial

We also measured body appreciation, functionality appreciation, appearance evaluation, physical functionality orientation, body checking, body image avoidance, appearance orientation, and eating disorder symptoms in a separate paper presenting the main outcomes of the randomized controlled IM FAB trial [27]. Using multilevel modeling, we found that the Functionality condition demonstrated significant improvements relative to the Active Comparator condition in Functionality Appreciation at T2 and T4, Body Appreciation at T2, and eating disorder symptoms at T2. Further, the Functionality condition exhibited significant improvements compared to the Assessment Only condition in functional appreciation at T2, appearance evaluation at T3 and T4, and body checking at T3. No other comparisons were consistently significantly different. Thus, functionality-focused ME in this study may be a useful a micro-intervention to improve positive body image. However, an in-depth of discussion of the intervention and of the main outcome findings are beyond the scope of this paper. The current paper sought to address the usability, feasibility, and engagement of the IM FAB procedure, which was delivered in person/on Zoom for mirror exposure and via text for the intervening weeks, as it was, and if it were to be delivered as a solely app-based intervention.”

Limitations of the sample

Only female undergraduates are included in the sample, making it a limited demographic. In the issue of generalizability, this constraint ought to be underlined further.

We agree with the Reviewer. We have made it clearer on the lack of generalizability of our sample, as follows:

“The results likely cannot be generalized to demographics aside from undergraduate women in the U.S. We could not evaluate how demographics factors affect the effectiveness of IM FAB program due to lack of statistical power to examine differences between multiple groups with low sample sizes.”

Were there differences in age, race, or body image issues? Demographic information would provide important background.

We thank the Reviewer for addressing this oversight on our part. We have added to our demographics Table (Table 1) the differences between the two conditions, none of which were statistically significantly different.

Level of feedback

It seems like the discussion of conflicting participant comments is not well-developed. Could the writers go into more detail regarding certain issues with involvement and accountability (such as a lack of reminders, a loss of social support, or a decline in motivation)?

We agree with the Reviewer. We have added more details regarding the reported issues with perceived presence in the room during the intervention and lack of intrinsic motivations, as follows:

“However, participants also noted they would have felt hesitant were others in the room (others were not in the room for the in-person mirror exposure, but may have been if participants completed the assignment in their dorm rooms, for example) and not being intrinsically motivated to engage in the IM FAB program if it were solely app-delivered and not part of an incentivized randomized control trial. It is noteworthy that some participants noted that they did not have body image concerns, but that it may be more effective for or motivating for those who do. As such, it is likely that some contextual factors exist (e.g., classes, roommates, time) that affected the perceived usability and feasibility of the intervention, even for those who were more intrinsically motivated.”

Were there any variations in the three weeks' engagement patterns (e.g., peak engagement, drop-off rates) that could help guide the design of the app?

We thank the Reviewer for the feedback. We agree that this would be helpful information. However, the way in which this study was conducted, as a lab-controlled RCT, where payments were predicated on completion of the requested questionnaires/texts, would be quite different from how users would use the app “in the wild.” So, we feel that the data we have probably do not usefully answer the question you have asked. We have added a bar chart showing user response rates (see Figure 1 below). Any frequency 6 or higher means that they completed all assigned text prompts and 61.24% of participants did complete all text prompts, many in great detail (text responses longer than a character limit set by the software company resulted in more than one cell per text response). We also added the following text:

We added the following to the engagement results:

“As an additional quantitative marker of engagement, at a Reviewer’s suggestion following preregistration, we examined completion rates. Participants were asked to respond to six text prompts. Each text above a pre-set character limit were sent as additional texts, such that six or more responses means participants completed all text prompts: 61.24% of participants completed all six responses. See Figure 1 for participants’ frequency of responding.”

Because of the way in which data were collected for text responses (from a software application) we have those data tied to participants’ user IDs in one dataset. Then we have responses with demographics and all questionnaire responses on another spreadsheet from our Qualtrics questionnaires. Thus, we do have sufficient data that we could assess whether demographic factors or condition related to response completion. However, (1) that would be an incredibly time-intensive process to match those two datasets up, and (2), more importantly, given our high response rates, and incentivized participation, I don’t expect it would suggest any meaningful differences here. If the Editor and Reviewers still think this is important, we are happy to take the time to do it (but the time required would exceed the typical review window provided). Given this is an important future direction and limitation of the current study, we have added this as well. 

“One important way in which this is likely the case is user engagement. Participants in our highly controlled randomized controlled trial were paid for completion of each part of the study. Thus, user engagement in real-world conditions would likely differ significantly. If IM FAB is developed into a mobile smartphone application, collecting engagement data without financially incentivizing participants to use the app is an important future direction to better understand and improve its user engagement over time.”

Delivery and fidelity of interventions

The organization and oversight of the mirror exposure sessions require further explanation. Did participants get any kind of guidance, or was adherence solely self-reported?

We thank the Reviewer for the feedback. For the mirror exposure sessions in the Functionality group, participants were guided through audio-recorded instructions on how to think and view their body in the mirror. For participants in the Active Comparator group, participants were only asked to view their body parts in the same order as the Functionality group. Research assistants were instructed to listen to ensure that participants were doing what was asked (e.g., getting up and out of chairs at the same time the audio recording was started, speaking to themselves when asked, etc.). We have added additional details to the manuscript as follows:

“Women in the Functionality condition received audio guided mirror exposure sessions on how to think about their body, focusing on their body’s physical, sensory, and creative capabilities. They were asked to do this in a room in a lab without a researcher present in the room with them. … in-person mirror exposure sessions. … In contrast, the Active Comparator condition received appearance-neutral mirror exposure, in which participants were only asked to examine body parts but not told how to think about those body parts. Like the Functionality condition, Active Comparator participants completed mirror exposure sessions listening to an audio recording in a private room in a laboratory.”

In a similar vein, how were journaling prompts sent and monitored? For app translation, it is essential to comprehend fidelity.

We added information about how texts were sent and monitored as follows:

“text message prompts that were scheduled in advance to be sent at three random times between 9 AM and 9 PM every other day in the two weeks between participants’ in-person mirror exposure sessions. These texts forwarded to the project director (DCW) for monitoring and were accessible via a password-protected website accessible to the research team.”

Directions for the future

Individualized features for upcoming app versions are mentioned in the end. It would be helpful to indicate which qualities the authors suggest or which participants asked for (e.g.,Reminders, gamification, progress monitoring, and personalization)

We thank the Reviewer for the suggestion. We have added more details on some of the individualized features that can possibly motivate users to use IM FAB more, as follows:

We added to our original content “Thus, any app developed would need to build in either gamification features or accountability features to help keep users engaged.” With the following recommendations:

“We recommend including features such as gamification, where participants can earn stars, connect with friends via their contacts, share stars with friends, create streaks for using the app each day, congratulate friends for completing streaks, allow them to refill a missed day, by completing an additional activity, show participants the testimonies from prior participants of the benefits of continuing the activities on the app every so often, and provide notifications to remind users to sign back in every so often, as examples. We used a unicorn as a “mascot” for the trial, so we envision that users could have a unicorn “mascot” that they start out with who is in grey scale, and they could see their unicorn become more vibrant and colorful and develop new features (e.g., wings, glitter, etc.) the more that users engage with the app.

And: “Future research should also explore motivational factors that would encourage more usage of IM FAB such as those noted previously (e.g., unicorn mascot who develops over time, connection with friends, keeping a streak by engaging, notification features, etc.), and with the ability to do exposure on the photo feature of the app or to do an at-home mirror exposure. Finally, studies can evaluate the extent of benefits that the inclusion of gratitude journaling will bring to participants with EDs and body image dissatisfaction, regardless of forms of interventions.”

Minor Remarks

Making a stronger connection between the possibility for IM FAB and digital technological deficiencies could tighten the introduction.

We thank the Reviewer for the suggestion. We have added the following to address this recommendation:

“IM FAB was created to address the lack of digital interventions for body image, especially among low-reach at-risk youths who do not have access to traditional interventions. We aimed to evaluate the usability of the intervention’s procedures and prompts, feasibility of use if it were translated into a fully digital intervention, as well as participant engagement, to best understand how the participants’ experiences can inform future DMHI development using the same treatment techniques.”

To make it easier for readers to follow findings across domains, think about rearranging the data to more clearly distinguish between usability, feasibility, and engagement.

We thank the Reviewer for the suggestion. In the original manuscript, we divided the domains by headings, and provided detailed explanations of each measure, including each item used in Table 2. We believe this is the most readable way to distinguish usability, feasibility, and engagement, and are not sure how to rearrange these to make the distinction clearer. However, we are also open to more specific suggestions on how to make the results and distinction between each outcome variable clearer. We did note one spot in the conclusions section, first paragraph where this was unclear, and clarified as follows:

“Similar to prior research, both the IM FAB Functionality condition and the Active Comparator condition demonstrated that usability (study’s procedures: mirror exposure and text prompts), feasibility for app-based delivery (degree to which they would have liked doing IM FAB on a smartphone app), and engagement (degree to which the mirror exposure and texts were interesting or engaging) were positively rated, supporting the importance of this initial step prior to intervention development in a digital format and widescale delivery.”

The abbreviation “IM FAB” is introduced suddenly. The name’s brief justification (such as “Function Appreciated Bodies”) ought to come first.

We thank the Reviewer for the suggestion. We have added a brief justification before the intervention name, as follows:

“The current study contributes to a growing implementation science literature by exploring the usability and acceptability of a digital intervention in which individuals appreciate the functions of their bodies in the mirror, called In the Mirror: Functional Appreciated Bodies (IM FAB).”

There is repetition in some of the wording (for example, “usability, feasibility, and engagement scores were excellent” might be more nuanced).

We thank the Reviewer for the feedback. We deleted the sentence that we believe the reviewer was referring to: “Overall, the usability, feasibility, and engagement scores were high” from the conclusions section.

This study’s standing in the literature would be improved by citations to relevant digital mental health interventions.

We thank the Reviewer for the suggestion. In the introduction section, we specifically cited three studies that assessed feasibility and usability of DMHIs, given that was the aim of the current paper. We have added relevant DMHI’s to the introduction to note that there are other ED and body image DMHIs, and noted shortcomings in the literature and public space, as follows:

“Although several ED DHMI’s have been developed and tested for initial efficacy, usability, feasibility, and engagement [15-19] these interventions are largely targeted towards those with active EDs rather than tailored towards reducing risk factors and preventing the development of EDs, save for two [20,21]. A recent review of all ED-related smartphone apps conducted in June 2021 noted 65 apps supporting ED treatment were on Google Play or the App store for iOS, with only 13 research articles describing and supporting four ED apps [22]. Additionally, the apps that have been tested are not always made publicly available soon after initial support for their efficacy, given the different timelines of academic vs consumer-facing products. Thus, there is both a need for in-novation in this space, but also for empirically-supported apps to be made publicly available, even at the same time as additional research is conducted and the product is refined. In this way, products with empirical support created by experts in may have greater prevalence in publicly available online marketplaces compared among available eating disorder and body image intervention DMHIs.”

Reviewer 2 Report

Comments and Suggestions for Authors

Dear authors,

After reading this manuscript, you should make the following comments and suggestions:

The participating women were randomly assigned to each condition. However, how were these participants initially selected? And what type of randomization was used in the assignment?

What is the purpose of the open-ended question asked after the UMUX questionnaire?

Cronbach's alpha assesses internal consistency within reliability, but it is not useful for assessing validity. To assess validity, other tests are needed to analyze face, content, criterion, or construct validity. Furthermore, the results of the Cronbach's alpha test are less than 0.7, so we must at least question the reliability of the UMUX questionnaire. Have the authors considered using the McDonald's omega test, which is more appropriate for analyzing the internal consistency of questionnaires with a response scale of fewer than 6 options?

It should be indicated who was responsible for conducting it, how it was conducted, and how the content validity of the feasibility questionnaire was analyzed.

I have the same suggestions and concerns regarding the participation questionnaire.

For a questionnaire to be reliable, it must be analyzed for internal consistency, homogeneity, and at least test-retest reliability. For a questionnaire to be valid, it must be analyzed for at least content validity and construct validity. Most of these aspects were not addressed in this research, so it can be said that these were ad hoc questionnaires pending adequate reliability and validity analysis. It cannot be stated so categorically that the questionnaires are valid and reliable.

I do not consider it appropriate to refer to a study to describe the procedure of this research. It should have been explained in detail in this manuscript.

More details should also be provided regarding the manipulation check questions: who designed them and how? How are they interpreted? What are the seven questions?

The data analysis is very brief. It should indicate what specific statistical tests were used to analyze the quantitative data. It should also clearly indicate how the qualitative data were analyzed.

The results should indicate the skewness of the items, in addition to the mean and standard deviation, to analyze whether the questionnaires are representative of the study population.

I observe in the results that some items have a very high standard deviation, greater than 1.5, so their variability within the study population is excessive, possibly indicating interpretation problems. Have the authors considered that these items should be modified or eliminated?

I believe the authors should have used the quantitative data obtained from the questionnaires to properly validate them in the study population. It is not enough to simply assess internal consistency, an essential requirement for verifying the validity and reliability of the results obtained. No conclusions can be drawn from the results obtained from these questionnaires, which were not validated for the study population.

I believe the qualitative responses should be categorized more clearly to facilitate reader understanding, even using a table.

I believe the conclusions should be clear and concise. The conclusions section includes a discussion and limitations that should be presented in a separate section before the conclusions.

I encourage the authors to properly validate the quantitative questionnaires with the data available to them in order to analyze the results obtained.

A reference to an article that may be helpful is attached:

García-Álvarez, J.M.; García-Sánchez, A.; Molina-Rodríguez, A.; Suárez-Cortés, M.; Díaz-Agea, J.L. Validation of the Group Environment Questionnaire (GEQ) in a Simulated Learning Environment. Nurs. Rep. 2025, 15, 154. https://doi.org/10.3390/nursrep15050154

Kind regards

Author Response

Reviewer 3

Dear authors,

After reading this manuscript, you should make the following comments and suggestions:

We thank Reviewer 3 for their time reviewing the manuscript and the detailed feedback. We have addressed each of the Reviewer’s comments in detail below.

The participating women were randomly assigned to each condition. However, how were these participants initially selected? And what type of randomization was used in the assignment?

We thank the Reviewer for the feedback. We have gone more in depth in our other paper, but we agree that this information should also be mentioned in this manuscript. We have added this information to the manuscript, as follows.

“Participants were not screened for ED or body image dissatisfaction. Participants were selected based on three inclusion criteria: (1) ≥ 18 years old; (2) self-identified as female; (3) had never participated in a cognitive-dissonance based body image group run on the same campus as one of the data collection sites [28].”

What is the purpose of the open-ended question asked after the UMUX questionnaire?

We thank the Reviewer for the clarifying question. We added the open-ended question to further understand how favorable participants were towards the intervention and the factors that our questionnaires could not capture. More specifically, we wanted to understand the aspects that participants believed enhanced their experience (e.g., wordings, comprehensibility), aspects that they believed would deter their experience (e.g., intrinsic motivation), and any aspect of the intervention that stood out to them. By knowing all of these factors, we had more guidance in suggesting what future research can look at and improve on this intervention. It would also help tailor app development in the future, should we receive grant funding to further this project.

Cronbach's alpha assesses internal consistency within reliability, but it is not useful for assessing validity. To assess validity, other tests are needed to analyze face, content, criterion, or construct validity. Furthermore, the results of the Cronbach's alpha test are less than 0.7, so we must at least question the reliability of the UMUX questionnaire. Have the authors considered using the McDonald's omega test, which is more appropriate for analyzing the internal consistency of questionnaires with a response scale of fewer than 6 options?

We have added omega in addition to the alpha already included. However, we note that because we have low item numbers, and a relatively small sample size, that our alpha estimates are higher than omega, and may be a better estimate (Orcan, 2023) and that alpha often represents an underestimate of actual reliability (Hoekstra et al., 2019).

We added the following to our limitations section: “Given the low number of items, especially for the UMUX, it is likely that coefficient alpha represents a more accurate estimate of the reliability of the scale than omega [34], and some researchers have noted concerns with the often used rule of thumb of a .70 cut-off [35] and that alpha is often an underestimate of reliability. Nonetheless, higher alpha and omega coefficients generally suggest higher internal consistency, and reliability is a necessary precondition for validity. We have yet to confirm the scales’ test-retest reliability, content validity and construct validity, all of which are crucial next steps, as is confirming its internal consistency.

It should be indicated who was responsible for conducting it, how it was conducted, and how the content validity of the feasibility questionnaire was analyzed.

The Reviewer’s concern here is unclear. We believe the Reviewer is asking about how we validated the feasibility and engagement scales (in the following point). We had not validated these measures. We noted this as a limitation and direction for future research.  

I have the same suggestions and concerns regarding the participation questionnaire.

For a questionnaire to be reliable, it must be analyzed for internal consistency, homogeneity, and at least test-retest reliability. For a questionnaire to be valid, it must be analyzed for at least content validity and construct validity. Most of these aspects were not addressed in this research, so it can be said that these were ad hoc questionnaires pending adequate reliability and validity analysis. It cannot be stated so categorically that the questionnaires are valid and reliable.

We agree with the Reviewer. We have added more information about the scale’s reliability and validity problem to the limitation section of our papers. Unfortunately, at the time of the development of these items, we were unable to find validated measures of feasibility that met our study’s needs or, similarly, a measure of engagement. We felt modifying an existing validated measure was likely to be the next best thing, in terms of likelihood of validity, though it’s at present undemonstrated.

I do not consider it appropriate to refer to a study to describe the procedure of this research. It should have been explained in detail in this manuscript.

We understand Reviewer 3’s concern. We were attempting to balance a number of concerns in referring the reader to the primary outcome paper for the bulk of the information on the procedures. The first concern is that if we completely reviewed the procedures and primary outcomes, as requested by other Reviewers, then the papers would be too overlapping, this paper would be too long, and it would feel too confusing in covering too many different kinds of outcomes. Because the purpose of this paper was only to review the usability, feasibility, and engagement in the IM FAB active intervention groups, and not to evaluate its efficacy, we opted to take the Reviewers’ feedback into account and provide greater detail about the procedure in the revision, but without providing too much detail that would render the papers too overlapping, and we felt that would obfuscate the purpose of the current manuscript.

We provided this level of information about the procedure, to flesh it out further in this revision in response to your and other Reviewers’ requests:

“Women in the Functionality condition received audio guided mirror exposure sessions on how to think about their body, focusing on their body’s physical, sensory, and creative capabilities. They were asked to do this in a room in a lab without a researcher present in the room with them. Additionally, Functionality participants received text message prompts that were scheduled in advance to be sent at three random times between 9 AM and 9 PM every other day in the two weeks between participants’ in-person mirror exposure sessions. These texts forwarded to the project director (DCW) for monitoring and were accessible via a password-protected website accessible to the research team. Texts asked participants to write about gratitude for different facets of their body’s functionality (e.g., mobility, strength, ability to connect with others) for Functionality participants. In contrast, the Active Comparator condition received appearance-neutral mirror exposure, in which participants were only asked to examine body parts but not told how to think about those body parts. Like the Functionality condition, Active Comparator participants completed mirror exposure sessions listening to an audio recording in a private room in a laboratory. Active Comparator participants received text prompts unrelated to body image intended to prompt gratitude for other parts of their lives (e.g., relationships, personal growth). We assessed participants’ functionality appreciation, body appreciation, eating disorder symptoms, body checking, body image avoidance, and multidimensional measures of body image. The randomized controlled trial demonstrated statistically significant improvements in body functionality appreciation and body image satisfaction for the Functionality condition compared to either the Active Comparator or an assessment-only control condition. For additional details about the IM FAB randomized controlled trial procedure and results, please see Costello et al. [27].”

More details should also be provided regarding the manipulation check questions: who designed them and how? How are they interpreted? What are the seven questions?

We thank the Reviewer for the clarifying question. The first author of this manuscript designed the manipulation check questions in consultation with research assistants who were working on the project. We have added the manipulation check questions to the supplementary materials. We note that manipulation check questions are more important for assessing the efficacy of between-group differences on primary outcome questions than they are for determining whether the tasks requested of the participants were usable, feasible for app-based delivery, and promoted user engagement, which are the questions we are addressing in this manuscript.

The data analysis is very brief. It should indicate what specific statistical tests were used to analyze the quantitative data. It should also clearly indicate how the qualitative data were analyzed.

We agree with the Reviewer. We have added more information on the statistical test we did with the quantitative data and the qualitative data in the results section.

The results should indicate the skewness of the items, in addition to the mean and standard deviation, to analyze whether the questionnaires are representative of the study population.

We have added skewness and kurtosis to Table 2, as requested. We note that all values except for the UMUX 4R item have acceptable skew and kurtosis values and that the scale averages have excellent skew and kurtosis values. We have added this to the limitations section.

I observe in the results that some items have a very high standard deviation, greater than 1.5, so their variability within the study population is excessive, possibly indicating interpretation problems. Have the authors considered that these items should be modified or eliminated?

As fellow psychometric/assessment afficionados, we appreciate the Reviewer’s attention to detail on the scale construction. We have noted the limitation of the item with higher SD, and added the scale totals to Table 2, for which the psychometric properties are much more sound.

I believe the authors should have used the quantitative data obtained from the questionnaires to properly validate them in the study population. It is not enough to simply assess internal consistency, an essential requirement for verifying the validity and reliability of the results obtained. No conclusions can be drawn from the results obtained from these questionnaires, which were not validated for the study population.

As fellow psychometric/assessment afficionados (DCW has developed and validated multiple measures and has more development/validation papers in process, with the appropriate methodology), we appreciate the Reviewer’s attention to detail on the scale construction, and do not disagree with your concerns.

We agree with the Reviewer that the scales are not properly validated yet. This was the first study conducting an RCT of the IM FAB protocol, for which we attempted to use only validated existing measures. Unfortunately, for a tertiary aim of our larger RCT project, assessing usability, feasibility in an app-based format for future research, and engagement in this IM FAB program, in particular, there was only the UMUX assessing usability, in terms of appropriate pre-existing validated measures. We did not find any engagement or feasibility questionnaires that would have worked to assess the constructs we needed to assess, as most scales examining these questions are for already-developed programs that are being tested as is, and that did not assess our specific intervention aspects (the mirror exposure and text prompts) for which we were evaluating engagement separately. Thus, we needed to develop our own questionnaires or modify pre-existing ones. We based them on the UMUX, our validated, existing questionnaire (which, of note, actually performed the worst psychometrically, despite prior validation), in our best attempt to rely on validated measures. Because this RCT was run with an early career grant with relatively limited funding, and those funding resources needed to be put into other parts of the study (largely participant and RA payment), it did not leave enough resources to separately validate these measures in advance of our study, as would have been ideal. This is one of those situations in research in which logistics and practicality by necessity override the “best case” and preferable methodology. Unfortunately, all research studies need to make sacrifices in this manner, due to limited time and monetary resources. We have noted the limitation of the questionnaire, and hope the Reviewers and editor feel similarly, that the study still retains value for publication, in particular in light of the inclusion of both qualitative and quantitative feedback on IM FAB’s usability, feasibility, and engagement.

Overall, we included the following to address the Reviewer’s concerns about validity and reliability: “Some of the items, in particular Feasibility item 3 (“I would probably not have actually done the mirror exposures at home if the study was delivered via an app.”) had a high standard deviation (2.13 on a 1-7 scale), indicating substantial variability within our sample, and high skewness and kurtosis in UMUX item 4. Nonetheless variability, skewness, and kurtosis were all acceptable for the total scales, suggesting appropriateness for analyses as a global scale [33]. Given the low number of items, especially for the UMUX, it is likely that coefficient alpha represents a more accurate estimate of the reliability of the scale than omega [34], and some researchers have noted concerns with the often used rule of thumb of a .70 cut-off [35] and that alpha is often an underestimate of reliability. Nonetheless, higher alpha and omega coefficients generally suggest higher internal consistency, and reliability is a necessary precondition for validity. We have yet to confirm the scales’ test-retest reliability, content validity and construct validity, all of which are crucial next steps, as is confirming its internal consistency.”

I believe the qualitative responses should be categorized more clearly to facilitate reader understanding, even using a table.

We thank the Reviewer for their suggestion. We have created a table in the supplementary materials for responses in each category: positive, negative, neutral.  We are unsure how to better categorize the results to make them clearer. Should the Reviewer have a specific suggestion, we are happy to implement that in a subsequent revision.

I believe the conclusions should be clear and concise. The conclusions section includes a discussion and limitations that should be presented in a separate section before the conclusions.

We have made the discussion and limitations into a separate sections, and appreciate the suggestion.

I encourage the authors to properly validate the quantitative questionnaires with the data available to them in order to analyze the results obtained.

We thank the Reviewer for the suggestion. However, the participant pool that we sampled on was in 2021 and we did not collect any other data from scales with similar constructs or scales that measure the same thing as the two scales we created, feasibility and engagement. Therefore, we apologize for our inability to further validate our quantitative questionnaires. We have provided additional information on the properties of the items and total scales, in addition to adding these concerns as limitations and future directions, as noted above.

A reference to an article that may be helpful is attached:

García-Álvarez, J.M.; García-Sánchez, A.; Molina-Rodríguez, A.; Suárez-Cortés, M.; Díaz-Agea, J.L. Validation of the Group Environment Questionnaire (GEQ) in a Simulated Learning Environment. Nurs. Rep. 202515, 154. https://doi.org/10.3390/nursrep15050154

We appreciate the suggested article. We understand the work necessary to validate a new measure; however, doing so for these measures that are very particular to the IM FAB Program for the feasibility and engagement scales, in particular, is both beyond the scope of this project at this time, and would have needed to occur prior to not following the data collection for this study. Additionally, it would not make sense to do validation questions a separate sample of individuals who had not actually experienced the IM FAB program, nor do we have resources to run two RCTs in order to validate these feasibility and engagement measures, unfortunately. Thus, it is not particularly feasible to validate these measures separately, and as noted above, was not in our budgetary or time constraints to be able to do so as part of the original study, conducted from 2018-2021. We have noted this as a considerable limitation and future direction in the current manuscript.

Round 2

Reviewer 1 Report

Comments and Suggestions for Authors

Nothng further. Just check English grammar

Author Response

We thank Reviewer 1 for their essential feedback and for attentively reading our manuscript. As suggested, we have thoroughly proofread our manuscript and corrected all identified grammatical errors.

Reviewer 2 Report

Comments and Suggestions for Authors

Dear Authors.

I consider that the manuscript has been sufficiently improved and I have no additional comments or suggestions.

Kind regards.

Author Response

We thank Reviewer 2 for their positive feedback and the time and effort they put into reviewing our manuscript.